# Conceptual structure coheres in human cognition but not in large language models

**Siddharth Suresh**[1], **Kushin Mukherjee**[1], **Xizheng Yu**[1],
**Wei-Chun Huang**[1], **Lisa Padua**[2] and **Timothy T. Rogers**[1]
[1]University of Wisconsin-Madison,[2]Albany State University
siddharth.suresh@wisc.edu

## Abstract

Neural network models of language have long been used as a tool for developing hypotheses about conceptual representation in the mind and brain. For many years, such use involved extracting vector-space representations of words and using distances among these to predict or understand human behavior in various semantic tasks. Contemporary large language models (LLMs), however, make it possible to interrogate the latent structure of conceptual representations using experimental methods nearly identical to those commonly used with human participants. The current work utilizes three common techniques borrowed from cognitive psychology to estimate and compare the structure of concepts in humans and a suite of LLMs. In humans, we show that conceptual structure is robust to differences in culture, language, and method of estimation. Structures estimated from LLM behavior, while individually fairly consistent with those estimated from human behavior, vary much more depending upon the particular task used to generate responses–across tasks, estimates of conceptual structure from the very same model cohere less with one another than do human structure estimates. These results highlight an important difference between contemporary LLMs and human cognition, with implications for understanding some fundamental limitations of contemporary machine language.

## 1 Introduction

Since Elman's pioneering work (Elman, 1990) showcasing the ability of neural networks to capture many aspects of human language processing (Rumelhart et al., 1986), such models have provided a useful tool, and sometimes a gadfly, for developing hypotheses about the cognitive and neural mechanisms that support language. When trained on a task that seems almost absurdly simplistic–continuous, sequential prediction of upcoming words in sentences–early models exhibited properties that upended received wisdom about what language is and how it works. They acquired internal representations that blended syntactic and semantic information, rather than keeping these separate as classic psycho-linguistics required. They handled grammatical dependencies, not by constructing syntactic structure trees, but by learning and exploiting temporal patterns in language. Perhaps most surprisingly, they illustrated that statistical structure latent in natural language could go a long way toward explaining how we acquire knowledge of semantic similarity relations among words. Because words with similar meanings tend to be encountered in similar linguistic contexts (Firth, 1957; Osgood, 1952; Harris, 1954), models that exploit contextual similarity when representing words come to express semantic relations between them. This enterprise of learning by predicting has persisted into the modern era of Transformer-based language models (Devlin et al., 2018; Brown et al., 2020a).

Though early work was limited in the nature and complexity of the language corpora used to train models (Elman, 1991; McClelland et al., 1990), these ideas spurred a variety of computational approaches that could be applied to large corpora of written text. Approaches such as latent semantic analysis (Deerwester et al., 1990) and skip-gram models (Mikolov et al., 2013a; Bojanowski et al., 2017), for instance, learn vector-space representations of words from overlap in their linguistic contexts, which turn out to capture a variety of semantic relationships amongst words, including some that are highly abstract (Grand et al., 2022; Elman, 2004; Lupyan and Lewis, 2019).

In all of this work, lexical-semantic representations are cast as static points in a high-dimensional vector space, either computed directly from estimates of word co-occurrence in large

text corpora (Deerwester et al., 1990; Burgess, 1998), or instantiated as the learned activation patterns arising in a neural network model trained on such corpora. To evaluate whether a given approach expresses semantic structure similar to that discerned by human participants, the experimenter typically compares the similarities between word vectors learned by a model to decisions or behaviors exhibited by participants in semantic tasks. For instance, LSA models were tested on synonym-judgment tasks drawn from a common standardized test of English language comprehension by comparing the cosine distance between the vectors corresponding to a target word and each of several option words, and having the model "choose" the option with the smallest distance (Landauer et al., 1998). The model was deemed successful because the choice computed in this way often aligned with the choices of native English speakers. Such a procedure was not just a useful way for assessing whether model representations are human-like—it was just about the *only* way to do so for this class of models.

In the era of large language models such as Open AI's GPT3 (Brown et al., 2020a), Meta's LLaMa family (Touvron et al., 2023; Taori et al., 2023), Google's FLAN (Wei et al., 2021), and many others (Zhang et al., 2022; Chowdhery et al., 2022; Hoffmann et al., 2022; Du et al., 2022), this has changed. Such models are many orders of magnitude larger than classical connectionist approaches, employ a range of architectural and training innovations, and are optimized on truly vast quantities of data—but nevertheless they operate on principles not dissimilar to those that Elman and others pioneered. That is, they exploit patterns of word co-occurrence in natural language to learn distributed, context-sensitive representations of linguistic meaning at multiple levels, and from these representations they generate probabilistic predictions about likely upcoming words. Current models generate plausible and grammatically well-formed responses, created by iteratively predicting what words are likely to come next and sampling from this distribution using various strategies (Wei et al., 2022; Wang et al., 2022; Yao et al., 2023). So plausible is the text that recent iterations like ChatGPT (Ouyang et al., 2022) can write essays sufficiently well to earn a A in an undergraduate setting (Elkins and Chun, 2020), pass many text-based licensing exams in

law and medicine (Newton and Xiromeriti, 2023; Choi et al., 2023; Kung et al., 2023) produce working Python code from a general description of the function (OpenAI, Year of the webpage, e.g., 2023), generate coherent explanations for a variety of phenomena, and answer factual questions with remarkable accuracy [1]. Even in the realm of failures, work has shown that the kinds of reasoning problems LLMs struggle with are often the same ones that humans tend to find difficult (Dasgupta et al., 2022). In short, if solely evaluated based on their generated text, such models *appear* to show several hallmarks of conceptual abilities that until recently were uniquely human.

These innovations allow cognitive scientists, for the first time, to measure and evaluate conceptual structure in a non-human system using precisely the same natural-language-based methods that we use to study human participants. Large language models can receive written instructions followed by a series of stimuli and generate interpretable, natural-language responses for each. The responses generated can be recorded and analyzed in precisely the same manner as responses generated by humans, and the results of such analyses can then be compared within and between humans and LLMs, as a means of understanding whether and how these intelligences differ.

The current paper uses this approach to understand similarities and differences in the way that lexical semantic representations are structured in humans vs LLMs, focusing on one remarkable aspect of human concepts–specifically, their *robustness*. As Rosch showed many years ago (Rosch, 1975, 1973), the same conceptual relations underlie behavior in a variety of tasks, from naming and categorization to feature-listing to similarity judgments to sorting. Similar conceptual relations can be observed across distinct languages and cultures (Thompson et al., 2020). Robustness is important because it allows for shared understanding and communication across cultures, over time, and through generations: Homer still speaks to us despite the astonishing differences between his world and ours, because many of the concepts that organized his world cohere with those that organize ours. Our goal was to assess whether conceptual structure

---

[1]While it is likely that GPT-3 has been trained on examples of many of these exams and tasks, that it can efficiently retrieve this knowledge when queried with natural language is nonetheless worth noting.

in contemporary LLMs is also coherent when evaluated using methods comparable to those employed with human participants, or whether human and LLM "mind" differ in this important regard.

To answer this question, we first measured the robustness of conceptual structure in humans by comparing estimates of such structure for a controlled set of concepts using three distinct behavioral methods – feature-listing, pairwise similarity ratings, and triadic similarity judgements – across two distinct groups – Dutch and North American – differing in culture and language. We then conducted the same behavioral experiments on LLMs, and evaluated (a) the degree to which estimated conceptual relations in the LLM accord with those observed in humans, and (b) whether humans and LLMs differ in the apparent robustness of such structure. We further compared the structures estimated from the LLM's overt patterns of behavior to those encoded in its internal representations, and also to semantic vectors extracted from two other common models in machine learning. In addition to simply demonstrating how methods from cognitive psychology can be used to better understand machine intelligence, the results point to an important difference between current state of the art LLMs and human conceptual representations.

## 2  Related work

In addition to many of the studies highlighted in the previous section, here we note prior efforts to model human semantics using NLP models. Many recent papers have evaluated ways in which LLMs are and are not humanlike in their patterns of behavior when performing tasks similar to those used in psychology–enough that, despite the relative youth of the technology, there has been a recent review summarising how LLMs can be used in psychology(Demszky et al., 2023), along with work highlighting cases where LLMs can replicate classical findings in the social psychology literature (Dillion et al., 2023). Looking further back, several authors have evaluated the semantic structure of learned word embeddings in static-vector spaces (Mikolov et al., 2013b; Marjieh et al., 2022b), while others have examined the semantic structure of more fine-grained text descriptions of concepts in language models capable of embedding sequences (Marjieh et al., 2022a). A few studies

have used models to generate lists of features and estimated semantic structure from feature overlap (Hansen and Hebart, 2022; Suresh et al., 2023; Mukherjee et al., 2023), or have asked models to produce explicit pairwise similarity ratings (Marjieh et al., 2023). While such work often compares aspects of machine and human behaviors, to our knowledge no prior study has evaluated the coherence of elicited structures across tasks within a given model, or between structures elicited from humans and machines using the same set of tasks.

## 3  Measuring Human Conceptual Structure

For both human and LLM experiments, we focused on a subset of 30 concepts (as shown in Table 1) taken from a large feature-norming study conducted at KU Leuven (De Deyne et al., 2008). The items were drawn from two broad categories–tools and reptiles/amphibians–selected because they span the living/nonliving divide and also possess internal conceptual structure. Additionally this dataset has been widely used and validated in the cognitive sciences.

To measure the robustness of conceptual structure in humans, we estimated similarities amongst the 30 items using 3 different tasks: (1) semantic feature listing and verification data collected from a Dutch-speaking Belgian population in the early 2000s, (2) triadic similarity-matching conducted in English in the US in 2022, and (3) Likert-scale pairwise similarity judgments collected in English in the US in 2023. The resulting datasets thus differ from each other in (1) the task used (feature generation vs triadic similarity judgments vs pairwise similarity ratings), (2) the language of instruction and production (Dutch vs English), and (3) the population from which the participants were recruited (Belgian students in early 2000's vs American MTurk workers in 2022/2023). The central question was how similar the resulting estimated structures are to one another, a metric we call *structure coherence*. If estimated conceptual similarities vary substantially with language, culture, or estimation method, the structural coherence between groups/methods will be relatively low; if such estimates are robust to these factors, it will be high. The comparison then provides a baseline against which to compare structural coherence in the LLM.

### 3.1 Methods

#### 3.1.1 Feature listing study

Data were taken from the Leuven feature-listing norms(De Deyne et al., 2008). In an initial *generation* phase, this study asked 1003 participants to list 10 semantic features for 6-10 different stimulus words which were were one of 295 (129 animals and 166 artifacts) concrete object concepts. The set of features produced across all items were tabulated into a 2600d feature vector. In a second *verification* phase, four independent raters considered each concept-feature pair and evaluated whether the feature was true of the concept. The final dataset thus contained a $C$ (concept) by $F$ (feature) matrix whose entries indicate how many of the four raters judged concept $C$ to have feature $F$. Note that this endeavour required the raters to judge hundreds of thousands of concept-property pairs.

From the full set of items, we selected 15 tools and 15 reptiles for use in this study (as shown in Table 1). We chose these categories because they express both broad, superordinate distinctions (living/nonliving) as well as finer-grained internal structure (e.g. snakes vs lizards vs crocodiles).

The raw feature vectors were binarized by converting all non-zero entries to 1, with the rationale that a given feature is potentially true of a concept if at least one rater judged it to be so. We then estimated the conceptual similarity relations amongst all pairs of items by taking the cosine distance between their binarized feature vectors, and reduced the space to three dimensions via classical multidimensional scaling (Kruskal and Wish, 1978). The resulting embedding expresses conceptual similarity amongst 30 concrete objects, as estimated via semantic feature listing and verification, in a study conducted in Dutch on a large group of students living in Belgium in the early 2010s.

#### 3.1.2 Triadic comparison study

As a second estimate of conceptual structure amongst the same 30 items, we conducted a triadic comparison or *triplet judgment* task in which participants must decide which of two option words is more similar in meaning to a third reference word. From many such judgments, ordinal embedding techniques (Jamieson et al., 2015; Hebart et al., 2022; Hornsby and Love, 2020) can be used to situate words within a low-dimensional space in which Euclidean distances between two words capture the probability that they will be selected as "most similar" relative to some arbitrary third word. Like feature-listing, triplet judgment studies can be conducted completely verbally, and so can be simulated using large language models.

*Participants* were 18 Amazon Mechanical Turk workers recruited using CloudResearch. Each participant provided informed consent in compliance with our Institutional IRB and was compensated for their time.

*Stimuli* were English translations of the 30 item names listed above, half reptiles and half tools.

*Procedure.* On each trial, participants viewed a target word displayed above two option words, and were instructed to choose via button press which of the two option words was most similar to the target in its meaning. Each participant completed 200 trials, with the triplet on each trial sampled randomly with uniform probability from the space of all possible triplets. The study yielded a total of 3600 judgments, an order of magnitude larger than the minimal needed to estimate an accurate 3D embedding from random sampling according to estimates of sample complexity in this task (Jamieson et al., 2015). Ninety percent of the judgments were used to find a 3D embedding in which pairwise Euclidean distances amongst words minimize the crowd-kernel triplet loss on the training set (Tamuz et al., 2011). The resulting embedding was then tested by assessing its accuracy in predicting human judgments on the held-out ten percent of data. The final embeddings predicted human decisions on held-out triplets with 75% accuracy, which matched the mean level of inter-subject agreement on this task.

#### 3.1.3 Pairwise similarity study

Our final estimate of conceptual structure relied on participants making similarity ratings between pairs of concepts from the set of 30 items using a standard 7 point Likert scale. Unlike the previous two methods which *implicitly* arrive at a measure of similarity between concepts, this approach elicits *explicit* numerical ratings of pairwise similarity. To account for the diversity in ratings between people, we had multiple participants rate the similarity between each concept pair in our dataset, with each participant seeing each pair in a different randomized order.

*Participants* were 10 MTurk workers recruited using CloudResearch. Each participant provided

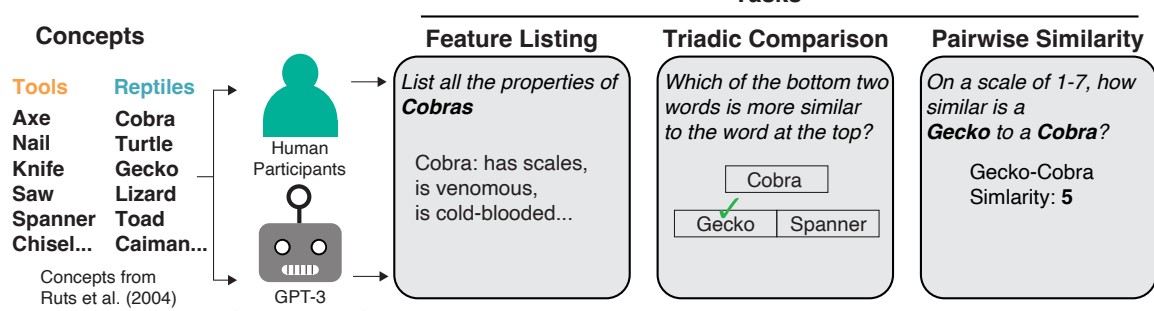

Figure 1: The three tasks used to estimate conceptual structure in both LLMs and Humans. The exact prompts used in our experiments with Humans and LLMs are shown in Table 4

informed consent in compliance with our Institutional IRB and was compensated for their time.

*Stimuli* were each of the 435 ($\binom{30}{2}$) possible pairs of the 30 tool and reptile concepts introduced in the earlier sections.

*Procedure.* On each trial of the experiment, participants were presented with a question of the form - *'How similar are these two things? {concept 1} and {concept 2}'* and were provided with a Likert scale below the question with the options — 1: Extremely dissimilar, 2: Very dissimilar, 3: Likely dissimilar, 4: Neutral, 5: Likely similar, 6: Very similar, 7: Extremely similar. On each trial {**concept 1**} and {**concept 2**} were randomly sampled from the set of 435 possible pairs of concepts and each participant completed 435 ratings trials rating each of the possible pairs.

### 3.2 Results

We found that the inter-rater reliability within the feature-listing and pairwise rating tasks were quite high ( $r = .81$ and $r = .98$ respectively). We could not compute a similar metric in a straightforward manner for the triadic judgement study because each participant was provided with a unique set of triplets. Figure 2 top row shows hierarchical cluster plots of the semantic embeddings from feature lists (left), the triadic comparison task (middle), and pairwise judgement task (right). Both embeddings strongly differentiate the living from nonliving items, and show comparatively little differentiation of subtypes within each category (though such subtypes are clearly apparent amongst the feature-listing embeddings). To assess the structural coherence among the three different embedding spaces, we computed the square of

the Procrustes correlation pairwise between the dissimilarity matrices for the 30 concepts derived from the three tasks (Gower, 1975). This metric, analogous to $r^2$, indicates the extent to which variations in pairwise distances from one matrix are reflected in the other. The metric yielded substantial values of 0.96, 0.84, and 0.72 when comparing representations from the feature-listing task to the triplet-judgement task, the feature-listing task to the pairwise comparison task, and the triplet task to the pairwise comparison task, respectively. All these values were significantly better than chance ($p < 0.001$), suggesting that in each comparison, distances in one space accounted for 96%, 84%, and 72% of the variation observed in the other.". Thus despite differences in language, task, and cultures, the three estimates of conceptual structure were well-aligned, suggesting that human conceptual representations of concrete objects are remarkably robust. We next consider whether the same is true of large language models.

## 4 Measuring Machine Conceptual Structure

In this section, we consider whether one of the most performant LLMs, OpenAI's **GPT 3**, expresses coherence in the structural organization of concepts when tested using the same methods used in the human behavioral experiments. Using the OpenAI API, we conducted the feature listing and verification task, triadic comparison task, and the pairwise similarity rating task on GPT 3. Given the recent deluge of open-source LLMs, we also tested **FLAN-T5 XXL**, and **FLAN-U2** on the triadic comparison and pairwise ratings tasks to see how they perform relative to larger closed models. Finally for completeness we also tested

| | Feature listing | Triadic comparisons | Pairwise ratings |
|---|---|---|---|
| **Humans** | | | |
| **GPT-3 (002)** | | | |
| **FLAN-XXL** | | | |

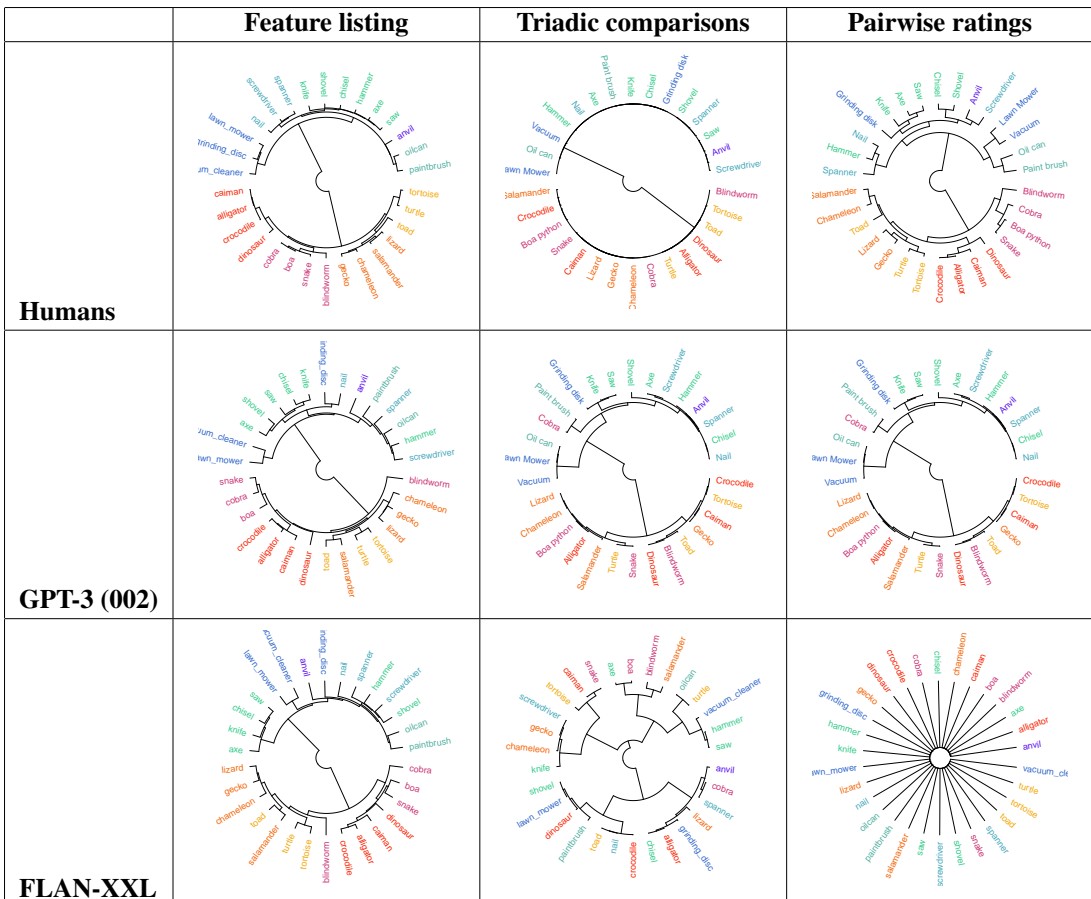

Figure 2: Organization of conceptual representations estimated from Humans, GPT-3, and FLAN-XXL using feature listing, triadic comparisons, and pairwise ratings. Grouping of concepts is based on hierarchical clustering of representations. Tools are shown in cooler colors and reptiles are in warmer colors.

the similarity between embeddings extracted from GPT 3, **word2vec**, and the language component of **CLIP**. While word2vec embeddings are a staple of NLP research, relatively fewer works have explored the structure of the language models that are jointly trained in the CLIP procedure.

After computing the similarity structure between concepts expressed by the NLP methods outlined above, we considered (a) how well these estimates aligned with structures estimated from human behaviors within each task, and (b) the structural coherence between the embeddings estimated via different methods from LLM behavior.

### 4.1 Methods

#### 4.1.1 Feature listing simulations

To simulate the feature-generation phase of the Leuven study, We queried GPT-3 with the prompt "List the features of a [concept]" and recorded the responses (see Table 3). The model was queried with a temperature of 0.7, meaning that responses were somewhat stochastic so that the model

produced different responses from repetitions of the same query. For each concept We repeated the process five times and tabulated all responses across these runs for each item. The responses were transcribed into features by breaking down phrases or sentence into constituent predicates; for instance, a response such as "a tiger is covered in stripes" was transcribed as "has stripes." Where phrases included modifiers, these were transcribed separately; for instance, a phrase like "has a long neck" was transcribed as two features, "has neck" and "has long neck." Finally, alternate wordings for the same property were treated as denoting a single feature; for instance, "its claws are sharp" and "has razor-sharp claws" would be coded as the two features "has claws" and "has sharp claws." We did not, however, collapse synonyms or otherwise reduce the feature set. This exercise generated a total of 580 unique features from the 30 items.

To simulate the feature verification phase of the Leuven study, we then asked GPT to decide, for each concept $C$ and feature $F$, whether the

concept possessed the feature. For instance, to assess whether the model "thinks" that alligators are ectothermic, we probed it with the prompt "In one word, Yes/No: Are alligators ectothermic?" (temperature 0). Note that this procedure requires the LLM to answer probes for every possible concept/feature pair–for instance, does an alligator have wheels? Does a car have a heart? etc. These responses were used to flesh out the original feature-listing matrix: every cell where the LLM affirmed that concept $C$ had feature $F$ was filled with a 1, and cells where the LLM responded "no" were filled with zeros. We refer to the resulting matrix as the *verified feature matrix*. Before the feature verification process, the concept by feature matrix was exceedingly sparse, containing 786 1's (associations) and 16614 0's (no associations). After the verification process, the concept by feature matrix contained 7845 1's and 9555 0's. Finally, we computed pairwise cosine distances between all items based on the verified feature vectors, and used classical multidimensional scaling to reduce these to three-dimensional embeddings, exactly comparable to the human study.

### 4.1.2 Triadic comparison simulations

To simulate triplet judgment, we used the prompt shown in Figure 1 for each triplet, using the exact same set of triplets employed across all participants in the human study. We recorded the model's response for each triplet (see Table 2) and from these data fit a 3D embedding using the same algorithm and settings as the human data. The resulting embedding predicted GPT-3 judgements for the held-out triplets at a 78 % accuracy, comparable to that observed in the human participants.

### 4.1.3 Pairwise similarity simulations

To simulate the pairwise similarity task, we used the prompt shown in Table 4 for all the possible pairs of concepts (435 ($\binom{30}{2}$)).

### 4.2 Results

Hierarchical cluster plots for embeddings generated from the LLM's feature lists, triadic judgements, and pairwise judgments are shown in the second and third rows of Figure 2, immediately below the corresponding plots from human data. Most approaches reliably separate living and nonliving things (although see the pairwise representations

for Flan-XXL to see a failure case). The verified feature lists additionally yield within-domain structure similar to that observed in human lists, with all items relatively similar to one another, and with some subcategory structure apparent (e.g. turtle/tortoise, snake, crocodile). within-domain structure estimated from triplet judgments, in contrast, looks very different.

These qualitative observations are borne out by the squared Procrustes correlations between different embedding spaces, shown in Figure 3. Similarities expressed in the space estimated from LLM verified feature lists capture 89% of the variance in distances estimated from human feature lists, 78% of the variance in those estimated from human triplet judgments, and 89% of the variance estimated from human pairwise similarities. Similarities expressed in the space estimated from LLM triplet judgments account for 82% of the variance in distances estimated from human feature lists, 80% of the variance in those estimated from human triplet judgments, and 69% of the variance estimated from human pairwise similarities. Finally similarities expressed in the space estimated from LLM Pairwise comparisons account for 32% of the variance in distances estimated from human feature lists, 14% of the variance in those estimated from human triplet judgments, and 51% of the variance estimated from human pairwise similarities. Similarities estimated from LLM pairwise comparisons, in contrast to those estimated from LLM feature lists and LLM triplets, account for less than half the variance in embeddings generated from human judgment. More interestingly, they account for less than half the variance in the embeddings generated from the LLM verified feature lists and LLM triplet judgements. Unlike the human embeddings, conceptual structures estimated from different behaviors in the very same model do not cohere very well with each other.

Figure 4 also shows the squared Procrustes correlations for semantic embeddings generated via several other approaches including (a) the raw (unverified) feature lists produced by GPT-3, (b) the word embedding vectors extracted from GPT-3's internal hidden unit activation patterns, (c) word embeddings from the popular word2vec approach, and (d) embeddings extracted from a CLIP model trained to connect images with their natural language descriptions. None of these

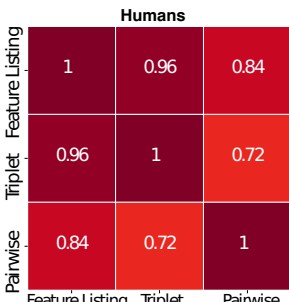 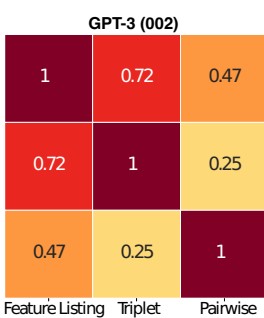 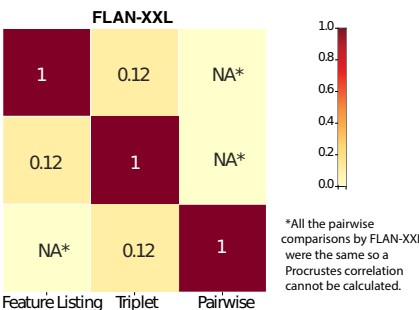

Figure 3: Matrices showing conceptual coherence across testing methods for Humans, GPT-3, and FLAN-XXL. The value in each cell corresponds to the squared Procustes pairwise correlation.

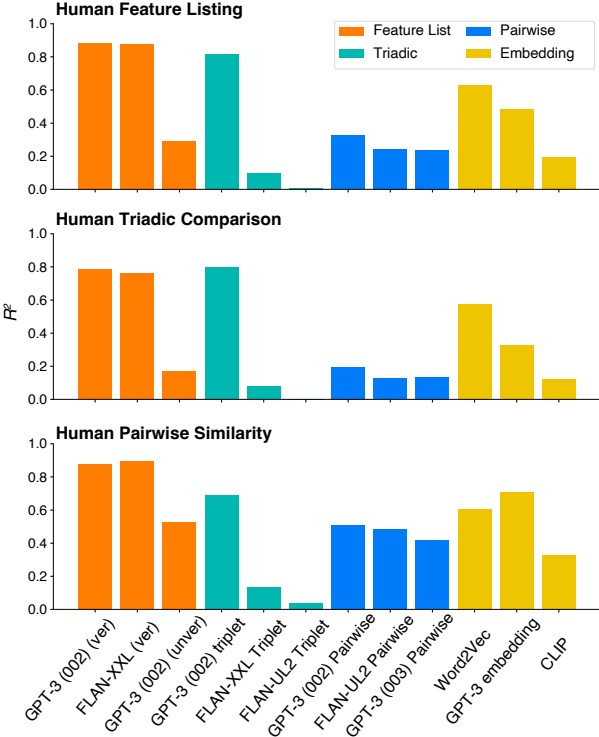

Figure 4: Human-machine alignment using human semantic representations estimated from feature listing (top), triadic comparisons (middle), and pairwise ratings (bottom). Heights of bars correspond to squared Procrustes correlations.

approaches accord with human-based embeddings as well as do the embeddings estimated from the LLM verified-feature lists, nor are the various structures particularly coherent with one another. No pair of LLM-estimated embeddings shows the degree of coherence observed between the estimates derived from human judgments.

In general, while LLMs vary in their degree of human alignment with respect to conceptual structure depending on the probing technique, the critical finding is that they are not coherent within-themselves across probing techniques. While

there might be ways to optimize human-machine conceptual alignment using in-context learning (Brown et al., 2020b; Chan et al., 2022) or specialized prompting strategies (Wei et al., 2022) to the extent that the community wants to deploy LLMs to interact with humans in natural language and use them as cognitive models, it is crucial to characterize how stable the models' concepts are without the use of specialized prompting.

## 5 Conclusion

In this study, we compared the conceptual structures of humans and LLMs using three cognitive tasks: a semantic feature-listing task, a triplet similarity judgement task, and a pairwise rating task. Our results showed that the conceptual representations generated from human judgments, despite being estimated from quite different tasks, in different languages, across different cultures, were remarkably coherent: similarities captured in one space accounted for 96% of the variance in the other. This suggests that the conceptual structures underlying human semantic cognition are remarkably robust to differences in language, cultural background, and the nature of the task at hand.

In contrast, embeddings obtained from analogous behaviors in LLMs differed depending upon on the task. While embeddings estimated from verified feature lists aligned moderately well with those estimated from human feature norms, those estimated from triplet judgments or from the raw (unverified) feature lists did not, nor did the two embedding spaces from the LLM cohere well with each other. Embedding spaces extracted directly from model hidden representations or from other common neural network techniques did not fare better: in most comparisons, distances captured by one model-derived embedding space accounted

for, at best, half the variance in any other. The sole exception was the space estimated from LLM-verified feature vectors, which cohered modestly well with embeddings taken directly from the GPT-3 embeddings obtained using the triplet task (72% of the variance) and the hidden layer (66% of variance)5.

While recent advances in prompting techniques including chain-of-thought prompting (Wei et al., 2022), self-consistence (Wang et al., 2022), and tree-of-thoughts (Yao et al., 2023) have been shown to improve performance in tasks with veridical solutions such as mathematical reasoning and knowledge retrieval, we highlight here through both direct and indirect tasks that the underlying *conceptual structure* learned by LLMs is brittle. We implemented chain-of-thought reasoning for some models and found that this led to LLM model representations being more aligned with human conceptual structure (Fig 6). However, the conceptual coherence within a model only increased for only some of the models but still nothing comparable to human conceptual robustness.

Together these results suggest an important difference between human cognition and current LLM models. Neuro-computational models of human semantic memory suggest that behavior across many different tasks is undergirded by a common conceptual "core" that is relatively insulated from variations arising from different contexts or tasks (Rogers et al., 2004; Jackson et al., 2021). In contrast, representations of word meanings in large language models depend essentially upon the broader linguistic context. Indeed, in transformer architectures like GPT-3, each word vector is computed as a weighted average of vectors from surrounding text, so it is unclear whether any word possesses meaning outside or independent of context. Because this is so, the latent structures organizing its overt behaviors may vary considerably depending upon the particular way the model's behavior is probed. That is, the LLM may not have a coherent conceptual "core" driving its behaviors, and for this reason, may organize its internal representations quite differently with changes to the task instruction or prompt. Context-sensitivity of this kind is precisely what grants such models their notable ability to simulate natural-seeming language, but this same capacity may render them ill-suited for understanding human conceptual representation.

# 6 Limitations

While there are benefits to studying the coherence of a constrained set of concepts, as we have done here, human semantic knowledge is vast and diverse and covers many domains beyond tools and reptiles. While it was reasonable to conduct our experiments on 30 concepts split across these domains both due to resource limitations and to limit the concept categories to those that are largely familiar to most people, a larger scale study on larger concept sets (Hebart et al., 2022; Devereux et al., 2014; McRae et al., 2005) might reveal a different degree of coherence in conceptual structure across probing methods in LLMs.

When conducting LLM simulations, we didn't employ any prompting technique like tree-of-thought(Yao et al., 2023), self-consistency(Wang et al., 2022), etc.. While we think that the purpose of this work is to highlight the fragility of the conceptual 'core' of models as measured by incoherent representations across tasks, it remains possible that representations might cohere to a greater extent using these techniques and might align closer with human representations.

Finally, Human semantic knowledge is the product of several sources of information including visual, tactile, and auditory properties of the concept. While LLMs can implicitly acquire knowledge about these modalities via the corpora they are trained on, they are nevertheless bereft of much of the knowledge that humans are exposed to that might help them organize concepts into a more coherent structure. In this view, difference in the degree in conceptual coherence between LLMs and humans should not be surprising.

# 7 Code

Here's the link to the repository containing the code and data we used to run our experiments.

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

# A Appendix

Table 1: Concepts used in the experiment

| Concepts | Categories |
|---|---|
| 15*Reptiles | Turtle |
| | Alligator |
| | Lizard |
| | Tortoise |
| | Cobra |
| | Snake |
| | Blindworm |
| | Gecko |
| | Boa python |
| | Toad |
| | Crocodile |
| | Chameleon |
| | Caiman |
| | Salamander |
| | Dinosaur |
| 15*Tools | Hammer |
| | Screwdriver |
| | Grinding disc |
| | Vacuum cleaner |
| | Spanner |
| | Lawn mower |
| | Axe |
| | Saw |
| | Knife |
| | Nail |
| | Chisel |
| | Shovel |
| | Anvil |
| | Oilcan |
| | Paint brush |

| Head | Option 1 | Option 2 | *FLAN-T5-XXL* | *FLAN-UL2* | *davinci-002* | *davinci-003* |
|---|---|---|---|---|---|---|
| Shovel | Alligator | Spanner | Alligator | Alligator | Spanner | Spanner |
| Anvil | Caiman | Tortoise | Tortoise | Caiman | Caiman | Caiman |
| Nail | Boa python | Snake | Snake | Snake | Boa Python | Boa python |
| Paint brush | Chisel | Toad | Chisel | Toad | Chisel | Chisel |
| Shovel | Caiman | Crocodile | Crocodile | Dangerous | Crocodile | Crocodile |

Table 2: Example responses to triplet judgement task by different models.

| Concepts | Prompt | *Features* |
| --- | --- | --- |
| Alligator | List all features of an Alligator. | have tail, can stay underwater, have tough skin ... |
| Anvil | List all features of an Anvil. | have wings, have hammer, have hole ... |
| Axe | List all features of an Axe. | have blade, can be used for chopping wood, are a tool ... |
| Blindworm | List all features of a Blindworm. | have no legs, can smell with tongue, are small ... |
| Boa python | List all features of a Boa python. | have pits, can be green, are dimorphic ... |
| Caiman | List all features of a Caiman. | have short body,can swim, are reptile ... |
| Chameleon | List all features of a Chameleon. | have tongue, can change color, are a good swimmer ... |
| Chisel | List all features of a Chisel. | have a blade, can cut material, are hand held ... |
| Cobra | List all features of a Cobra. | have long body, can inject venom, are carnivorous ... |
| Crocodile | List all features of a Crocodile. | have teeth, can breathe air, are a swimmer ... |
| Dinosaur | List all features of a Dinosaur. | have claws, can lay eggs, are reptiles ... |
| Gecko | List all features of a Gecko. | have tail, can stick to surfaces, are nocturnal ... |
| Grinding disk | List all features of a Grinding disk. | have diameter, can be used for grinding. are abrasive ... |
| Hammer | List all features of a Hammer. | have handle, can be used for pounding nails, are a tool ... |
| Knife | List all features of a Knife. | have blade , can cut things, are sharp ... |
| Lawn Mower | List all features of a Lawn Mower. | have engine, can mulch, are powered by gas ... |
| Lizard | List all features of a Lizard. | have legs,can climb, are carnivores ... |
| Nail | List all features of a Nail. | have covering, have point, are metal ... |
| Oil can | List all features of an Oil can. | have a spout, can pour, are used to hold oil ... |
| Paint brush | List all features of a Paint brush. | have handle, can be used for painting, are a tool ... |
| Salamander | List all features of a Salamander. | have tail, can be a variety of colors, are an amphibian ... |
| Saw | List all features of a Saw. | have handle, can be used to cut lumber, are a tool ... |
| Screwdriver | List all features of a Screwdriver. | have a handle, have a tip, have a cap ... |
| Shovel | List all features of a Shovel. | have point, have blade, have handle ... |
| Snake | List all features of a Snake. | have long body, can be dangerous to humans, are carnivorous ... |
| Spanner | List all features of a Spanner. | have different tips, can grip a bolt, are a tool ... |
| Toad | List all features of a Toad. | have glands, can jump, are good jumpers ... |
| Tortoise | List all features of a Tortoise. | have shell, can live long, are cold blooded ... |
| Turtle | List all features of a Turtle. | have shell, can swim, are reptiles ... |
| Vacuum | List all features of a Vacuum. | have nozzle, can suck up dirt, have filter ... |

Table 3: Examples of features produced by GPT-3.

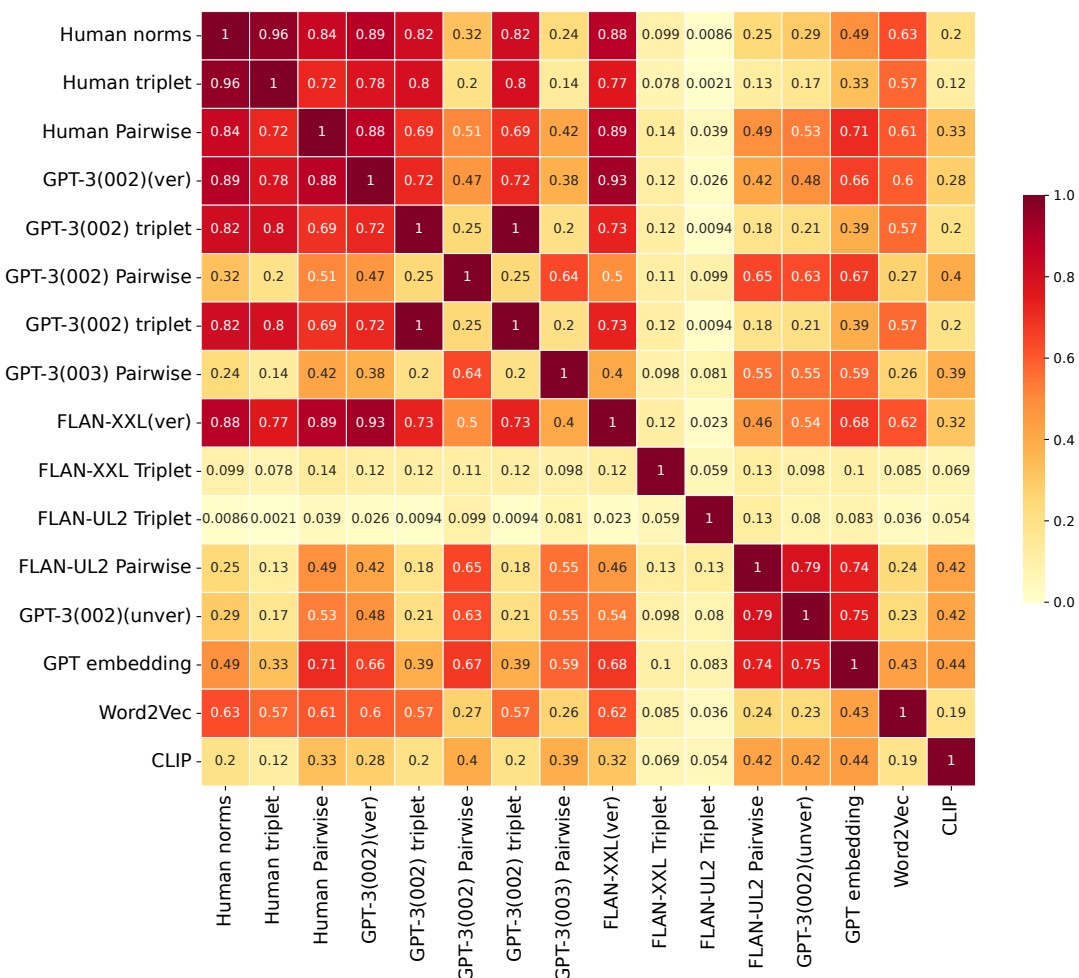

Figure 5: A correlation matrix showing square of the Procrustes correlations between semantic representations estimated from different models across different tasks.

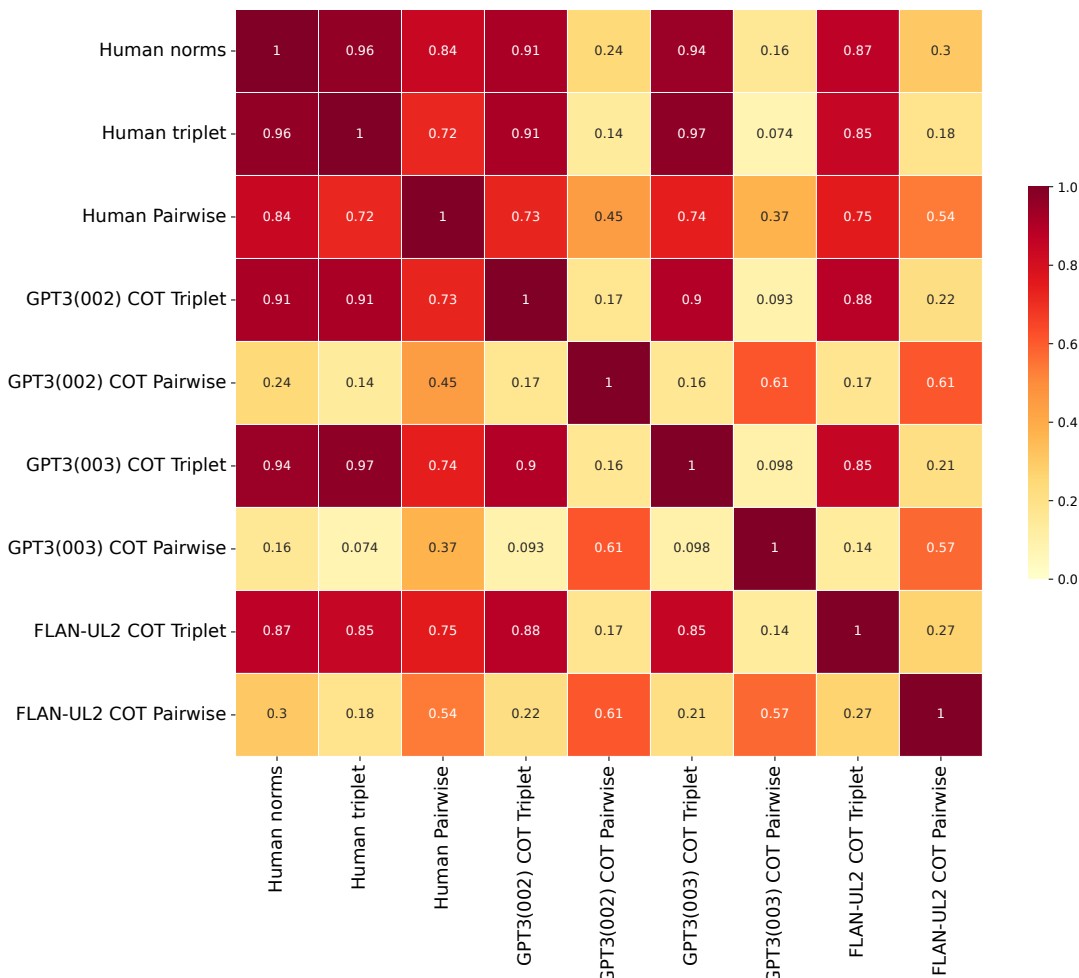

Figure 6: A correlation matrix showing square of the Procrustes correlations between semantic representations estimated from different models across different tasks using only Chain of Thought prompting.

| | **Humans** | **LLMs** |
|---|---|---|
| Feature generation | This bundle contains up to 10 sheets with a word written on top of the page. We would like you to write down preferably 10 features underneath the word. Try to give different sorts of features, such as, for example, physical or perceptual features (what it looks like, how it smells, how it tastes, ...), functional features (what it is used for, when and where it is used, ...), background information (where it comes from, some historical facts, ...), etc. ((De Deyne et al., 2008)) | List all the properties of {concept1} |
| Feature verification | The participants were instructed to judge, for every feature-exemplar pair, whether the feature characterizes the exemplar, and to write down a 1 or a 0 in the corresponding matrix entry. (De Deyne et al., 2008) | In one word, Yes/No : Are {concept1} {property1} (or) In one word, Yes/No : Do {concept1} have {property1}/ |
| Triplet task | Which of the bottom two words is more similar to the word at the top? | Answer using only one word - {concept1} or {concept2} and not {anchor}. Which is more similar in meaning to {anchor}?" |
| Pairwise similarity | On a scale of 1-7, how similar is a {concept1} to a {concept2}? | Answer with only one number from 1 to 7, considering 1 as 'extremely dissimilar', 2 as 'very dissimilar', 3 as 'likely dissimilar', 4 as 'neutral', 5 as 'likely similar', 6 as 'very similar', and 7 as 'extremely similar': How similar is {concept1} and {concept2}? |

Table 4: Prompts used across the three tasks for humans and LLMs