# OpenReview forum: "Conceptual structure coheres in human cognition but not in large language models"
_EMNLP/2023/Conference — EMNLP 2023 Main_

### Official Review · Reviewer_P15m · 2023-08-03

**Soundness:** 5

**Excitement:**

4: Strong: This paper deepens the understanding of some phenomenon or lowers the barriers to an existing research direction.

**Missing References:**

I was surprised not see the work by Lenci, Sahlgren and colleagues on the analysis of LLMs' embeddings discussed in the current work, as it also tests several models on a host of semantic tasks and reports interesting patterns about the degree to which underlying embedding spaces cohere or not.
https://arxiv.org/abs/2105.09825

**Paper Topic And Main Contributions:**

The paper presents an assessment of the robustness of conceptual representations in humans and LLMs. Humans and models are probed on the same three tasks (feature listing, triplet similarity and explicit similarity ratings), allowing to assess how coherent underlying conceptual spaces are. For humans, different tasks are administered to different populations in terms of culture and language (Dutch and English), adding to the credibility of the high robustness exhibited by human-derived conceptual spaces. The work also goes beyond probing LLMs by prompting and analyses underlying embedding spaces, also including word2vec and CLIP, which offers a first idea of how including visual information in the training regime affects derived representations.

Results show a much stronger coherence in human conceptual spaces, where one can be used to predict another even when the population who provided the ratings differed. For LLMs, figures vary a lot: feature verification and triplet similarity performed by GPT3 offer a good correlation with human tasks (judging from Figure 5, all numbers are above .8 - but explicit similarity ratings on Likert scales do a poor job. FLAN-XXL (ver) also does a decent job at capturing human conceptual spaces, but other tasks performed by the same model do a poor job with respect to the same FLAN XXL (ver) and humans, underscoring the point made by the author that model-based spaces are often brittle and task dependent unlike humans'. Other models lag behind considerably.

----------------------------------------------------------------
I appreciate all the responses. I haven't increased my 4 on Excitement to 5 as I don't think the paper is transformative; I have raised the score on reproducibility considering the extra details provided in the rebuttals. That said, I do think this paper should feature in the conference.

The work is methodologically solid and insightful, offering a well-rounded evaluation of consistency in conceptual spaces in terms of tasks, variation in human populations, breadth scope of models probed, procedure, and presentation of results. The number of concepts and the sample size of raters are a bit concerning, as I will expand below, but all in all I think this paper should have a place in the main conference.

**Questions For The Authors:**

Why do you map feature spaces to three dimensions? Unless I'm missing something, all evaluations would work in higher dimensional spaces as well, and this dimensionality reduction may get rid of signal. Could this be justified better?

It seems like participants did not flinch, but a screwdriver may be a drink and the translation will inevitably get rid of some polysemy while potentially introducing some which was not there. Did you handle this? If so, how? If not, can you comment on how this may impact your results? Participants saw all stimuli and may have understood polysemous terms in the intended meaning because of surrounding concepts, but LLMs may not since, if I understand correctly, you presented each prompt separately, making it impossible for the model to use preceding trials as context. In a similar fashion,

When you sample 90% of the responses for training and 10 for testing, how was the sampling performed? fully random? stratified on something? You mention 75% accuracy in out-of-sample prediction: could you provide a stupid baseline for comparison to better appreciate the figure?

I'm slightly concerned but mostly interested in possible order effects in the human data for pairwise similarity ratings. Given the very closed domain and the repetition of concepts in different pairs, is it possible that the way people interpreted the Likert scale changed as they saw more pairs? to exemplify: if I see the pair boa-crocodile first, I may choose a 6, indicating they are very similar. However, if I then see the pair crocodile-caiman I may also give a 6 but if the two trials are presented in the opposite order crocodile-boa would look less similar and probably elicit a 5. Considering that all raters rated all pairs in different order, I think it would be informative to investigate whether the order played a role. I don't have strong intuitions about how this analysis should be carried out, but I'd appreciate seeing it in an appendix to better situate the results.

How does the verification phase work? Does GPT-3 hallucinate in front of crocodiles having blades or not? did you check that the verification worked or did you trust it as such?


-------------------------------------------------------------
Thanks for the thorough and insightful answers!

**Reasons To Accept:**

The paper is sound, timely, accessible, presents a well-rounded comparison in terms of tasks and models, the way data were obtained from participants adds to making this paper interesting as it effectively taps into possible sources of variation. Results are interesting and highlight interesting differences between humans and models which should be further investigated.

**Reasons To Reject:**

I have only one real possible reason why I may not fully trust the results: human participants did the triplet similarity responding to a prompt asking to decide which of two words is more similar to the target, with no further qualifiers. By my understanding, the model was prompted with the same text. Is there a chance that some models did not interpret similar as semantically similar but rather, e.g., similar from a word from point of view? would results differ if the prompt was more specific about meaning relations? I agree that if we have to change the task specification for a model to perform well whereas humans are flexible enough to fill in the ambiguity and do the task it's not really good for the models, but Turner's are skilled at taking experiments whereas LLMs may not. The same concern applies to the pairwise similarity: can it be confirmed that the models interpret the 'similar' in the prompt as pertaining to lexical semantics?

The sample size, both in terms of number of concepts and semantic categories, as well as in terms of raters sampled is limited. I agree the choice of tools and reptiles captures something interesting, but I'm more worried about showing all stimuli to only a handful of raters, which may bring in very specific biases. Could you show what the agreement between raters on a same task is, maybe in an appendix?

-----------------------------------------
I think agreement puts many concerns to rest, thanks!

**Reproducibility:**

5: Could easily reproduce the results.

**Reviewer Confidence:**

5: Positive that my evaluation is correct. I read the paper very carefully and I am very familiar with related work.

**Typos Grammar Style And Presentation Improvements:**

The paper is very well written and clear in presenting results. I only find Figure 4 sub-optimal and somewhat hard to read at a glance, but at the same time I don't have any good advice on how to improve it. A few typos I'm sure the authors will catch upon reading the paper again (We capitalised in the middle of sentences, a couple of reduplications, nothing major).

In Figure 5 one row is duplicated (as evidenced by correlations of 1 in off-diagonal cells) and it's not fully clear what the difference between GPT-3 (002) pairwise and GPT-3 (003)-pairwise is.

---

> ### Author Rebuttal · Authors · 2023-08-29
>
> We thank the reviewer for their positive comments and for the thoughtful critiques. Below we respond to them in a point-by-point manner-
>
> >I have only one real possible reason why I may not fully trust the results: human participants did the triplet similarity responding to a prompt asking to decide which of two words is more similar to the target, with no further qualifiers. By my understanding, the model was prompted with the same text. Is there a chance that some models did not interpret similar as semantically similar but rather, e.g., similar from a word from point of view? would results differ if the prompt was more specific about meaning relations? I agree that if we have to change the task specification for a model to perform well whereas humans are flexible enough to fill in the ambiguity and do the task it's not really good for the models, but Turner's are skilled at taking experiments whereas LLMs may not. The same concern applies to the pairwise similarity: can it be confirmed that the models interpret the 'similar' in the prompt as pertaining to lexical semantics?
>
> We agree that it is hard to understand if the model is paying attention to the semantic similarity or the word similarity. In our triplet and pairwise task, we asked the model to make a judgment based on the ‘meaning’ of the words as shown by the prompt table in response to reviewer DWHV. To make this more concrete, we ran a new set of experiments (on triplet judgment and pairwise similarity )where we asked Flan-xxl and gpt-3(002) to make a judgment based on the ‘semantic meaning’ of the words. We find that in some cases, it decreases coherence but in some, it increases coherence. In the cases where the model coherence increases, it is not close to the coherence observed in humans.
>
> >The sample size, both in terms of number of concepts and semantic categories, as well as in terms of raters sampled is limited. I agree the choice of tools and reptiles captures something interesting, but I'm more worried about showing all stimuli to only a handful of raters, which may bring in very specific biases. Could you show what the agreement between raters on a same task is, maybe in an appendix?
>
> We computed the inter-rater reliability for the feature listing tasks and the pairwise tasks and found them to be 0.81, 0.98 respectively.
>
>
>
>
> >Why do you map feature spaces to three dimensions? Unless I'm missing something, all evaluations would work in higher dimensional spaces as well, and this dimensionality reduction may get rid of signal. Could this be justified better?
>
> Thank you for prompting us to clarify our analysis decisions. When fitting triplet embeddings, we often want to fit an embedding that is compact yet expressive. A 3D embedding for our data predicted 77% of the held-out data which is comparable rates with other studies in this domain. Since we chose to conduct a Procrustes correlation analysis, all the embedding dimensions had to be the same, hence the choice of 3 (the smallest dimensionality).
> We agree that all evaluations would work in higher dimensional spaces as well. We re-did the analysis in 30d (since the distance matrix for 30 concepts is 30*30), and the Procrustes correlations were slightly higher. However, it didn’t change any of the conclusions of the paper and we observe a similar trend as before.
>
>
> >It seems like participants did not flinch, but a screwdriver may be a drink and the translation will inevitably get rid of some polysemy while potentially introducing some which was not there. Did you handle this? If so, how? If not, can you comment on how this may impact your results? Participants saw all stimuli and may have understood polysemous terms in the intended meaning because of surrounding concepts, but LLMs may not since, if I understand correctly, you presented each prompt separately, making it impossible for the model to use preceding trials as context.
>
> Thank you for this thought-provoking comment! We posit that if humans are greatly sensitive to polysemy in the concepts used, this would lead to less coherent representations across testing modalities (since, e.g., the features listed are for the tool screwdriver, but in the triplet task and similarity task participants are free to interpret whichever meaning they like). But this is not the case, and representations are quite coherent in humans. It is only in models where the polysemous information carried by a word might be negatively impacting coherence, another point of difference between humans and overly context sensitive LLMs. In terms of participants seeing all the stimuli and the models only seeing individual stimuli, we randomized the order in which stimuli were presented such that if there were any order effects, they would be mitigated when averaging across participants.
> We agree with the point that participants have memory and can use preceding trials as context whereas LLMs cannot. Future studies can explore how these results change with in-context learning[1] by providing few-shot examples in the prompt.
>
> References:
> [1]​​Dong, Q., Li, L., Dai, D., Zheng, C., Wu, Z., Chang, B., ... & Sui, Z. (2022). A survey for in-context learning. arXiv preprint arXiv:2301.00234.
>
>
> >When you sample 90% of the responses for training and 10 for testing, how was the sampling performed? fully random? stratified on something?
>
> Sampling was performed randomly with uniform probability and without replacement as specified in prior work [1].
>
> References:
> [1] Sievert, S., Nowak, R., & Rogers, T. (2023). Efficiently Learning Relative Similarity Embeddings with Crowdsourcing. Journal of Open Source Software, 8(84), 4517.
>
>
> >You mention 75% accuracy in out-of-sample prediction: could you provide a stupid baseline for comparison to better appreciate the figure?
>
> Chance in this case would be 50%, since in a given triplet one has a 50/50 chance of choosing the ‘correct’ response.
>
>
> >I'm slightly concerned but mostly interested in possible order effects in the human data for pairwise similarity ratings. Given the very closed domain and the repetition of concepts in different pairs, is it possible that the way people interpreted the Likert scale changed as they saw more pairs? to exemplify: if I see the pair boa-crocodile first, I may choose a 6, indicating they are very similar. However, if I then see the pair crocodile-caiman I may also give a 6 but if the two trials are presented in the opposite order crocodile-boa would look less similar and probably elicit a 5. Considering that all raters rated all pairs in different order, I think it would be informative to investigate whether the order played a role. I don't have strong intuitions about how this analysis should be carried out, but I'd appreciate seeing it in an appendix to better situate the results.
>
> Thank you for allowing us to explain the rationale behind our experimental design in further detail. Order effects do exist and are prevalent in many psychological studies. Here, our guard against this is via randomization — that is, randomly shuffling the order in which pairs/triplets were presented to participants and averaging data across participants eliminates any reliable effect of order. As noted in a previous response, if any order effects were to ‘seep through’ in our human studies, this would actually cause us to underestimate the degree of conceptual coherence across tasks. We computed the inter-rater reliability for the feature listing tasks and the pairwise tasks and found them to be 0.81, 0.98 respectively.
>
> >How does the verification phase work? Does GPT-3 hallucinate in front of crocodiles having blades or not? did you check that the verification worked or did you trust it as such?
>
> We describe the prompting strategy for feature verification in response to reviewer DWHV.
> There is no ground truth to check if the process worked. If a model hallucinates and states that crocodiles have blades, we note this as the response chosen by the model. We analyzed the feature verification of Flan-XXL in prior work and developed[1] a method to increase the accuracy of feature verification [2].
>
> >I was surprised not see the work by Lenci, Sahlgren and colleagues on the analysis of LLMs' embeddings discussed in the current work, as it also tests several models on a host of semantic tasks and reports interesting patterns about the degree to which underlying embedding spaces cohere or not. https://arxiv.org/abs/2105.09825
>
> We apologize for missing this very pertinent work and thank the reviewer for bringing it to our attention! We will plan to add this reference and a sentence summarizing the findings in our related work section in our revision if the paper were to be accepted.
>
> References:
> [1] Suresh, S., Mukherjee, K., & Rogers, T. T. (2023). Semantic Feature Verification in FLAN-T5. arXiv preprint arXiv:2304.05591.
> [2]Mukherjee, K., Suresh, S., & Rogers, T. T. (2023). Human-machine cooperation for semantic feature listing. arXiv preprint arXiv:2304.05012.

---

### Official Review · Reviewer_qZ7X · 2023-08-04

**Soundness:** 5

**Excitement:**

4: Strong: This paper deepens the understanding of some phenomenon or lowers the barriers to an existing research direction.

**Paper Topic And Main Contributions:**

This work compared conceptual structure in humans with large language models. At issue was how coherent conceptual structure was across different tasks (and for humans different languages). Three paradigms were employed: (i) a feature listing task, (ii) a triadic similarity-matching task, and (iii) a Likert-scale pairwise similarity task. While human responses are consistent across tasks, languages, and populations, neural languages models are not.

The main contribution of this work is a careful investigation of the conceptual knowledge of a range of neural models. If the materials from this paper are released publicly upon publication, the paper also contributes a paradigm for evaluating models along with some human baseline data.


**Questions For The Authors:**

A. Have you tried running the Dutch experiments with GPT-3? It is unclear exactly what languages the model has been exposed to, but it might be an additional point of comparison to see how consistent measures derived from the same model in other languages are (note, I understand that this isn’t possible for some of the other models investigated, so I understand why it isn’t included in the paper).

**Reasons To Accept:**

This paper is great and enjoyable to read. The argument is clear, the experiments are well described and motivated, and the results are engaging. The work highlights the brittleness of conceptual knowledge in neural models, which will (hopefully) spur researchers to explore new modeling techniques to bridge the gap between humans and state-of-the-art models.

**Reasons To Reject:**

There are two (minor) issues in the paper: (i) a framing problem around the (actual) capacities of current models, and (ii) sparse discussion of the validity of the model comparisons. These are not reasons to reject, but there is no natural place in the review form to put these, so I include them here. Ultimately, I believe these two issues are easily addressable in revisions and that the paper should be accepted.

With respect to framing, the introduction of the paper asserts a few claims about the abilities of models like Chat GPT that are unsupported textually and/or difficult to substantiate (and thus prone to unintended misreadings). For example, lines 132-136 list a variety of abilities of ChatGPT, including the claim that the model “can write essays sufficient to earn a B in a typical college class”. No citation is given for these claims. Additionally, it is claimed that the model demonstrates “hallmarks of conceptual abilities that until recently were uniquely human”. This can be read as asserting that models have these conceptual abilities, which is at-issue in the paper. Finally, models are deemed intelligent, which is not substantiated. Again, these are minor.

With respect to model comparisons, there is an implicit move to treat different models (sometimes quite different, e.g., word2vec vs. GPT-3.5) as analogous to different human participants (or populations). It is possible, that only certain models produce conceptual structures which are comparable, and thus, this comparison is problematic. The results in the paper about the lack of consistency across tasks shows that even for a single model, they behave in non-human-like ways. Nonetheless, it could be beneficial to note this in the paper.


**Reproducibility:**

5: Could easily reproduce the results.

**Reviewer Confidence:**

4: Quite sure. I tried to check the important points carefully. It's unlikely, though conceivable, that I missed something that should affect my ratings.

**Typos Grammar Style And Presentation Improvements:**

Line 279: norms( --> norms (

Line 601: variance)5. Presumably 5 is a typo

Line 663: Finally, Human --> Finally, human

---

> ### Author Rebuttal · Authors · 2023-08-29
>
> We thank the reviewer for taking the time to read our paper and for their encouraging comments. Below, we address the concerns raised in a point-by-point manner.
>
> >With respect to framing, the introduction of the paper asserts a few claims about the abilities of models like Chat GPT that are unsupported textually and/or difficult to substantiate (and thus prone to unintended misreadings). For example, lines 132-136 list a variety of abilities of ChatGPT, including the claim that the model “can write essays sufficient to earn a B in a typical college class”. No citation is given for these claims. Additionally, it is claimed that the model demonstrates “hallmarks of conceptual abilities that until recently were uniquely human”. This can be read as asserting that models have these conceptual abilities, which is at-issue in the paper. Finally, models are deemed intelligent, which is not substantiated. Again, these are minor.
>
> Thank you for prompting us to be more careful in the claims we make and in our choice of wording. Here, we note some of the references that would help support our claims. The claims range from documented news articles - ChatGPT scoring Bs in essays: https://www.wbur.org/news/2023/07/26/harvard-student-chat-gpt-experiment-maya-bodnick— to further evidences from earlier last year documenting the ability of off-the-shelf chatGPT to pass multiple-choice exams [1], medical licencing exams [2], and the exams relating to the law [3]. Of course, this is because chatGPT might be trained on corpora containing materials from these exams, but it nevertheless remains unchanged that within these domains its behavior is impressive. We would add these references in addition to the caveat regarding the training data as mentioned above, if this paper were to be accepted.
>
> Regarding our claim that chatGPT `demonstrates “hallmarks of conceptual abilities that until recently were uniquely human”`, we completely agree that this sentence could be misleading. Our point was that on the surface, it appears as though these models have conceptual abilities if we were to simply judge the model based on the generated text for most normal conversation scenarios. However, as we demonstrate with our experiments in this paper, this is not the case and GPT-like models do not in fact have a conceptual core.
>
> To our knowledge, we do not refer to the models as ‘intelligent’ anywhere. If the reviewer is referring to the sentence on line 156 ‘as a means of understanding whether and how these intelligences differ’, we use the term intelligence liberally to refer to any system that is capable of generating output given a query. We can relax this and refer to the GPT models as ‘agent’ or just be direct and say the ‘difference between humans and Transformer models’, if the reviewer believes it to be appropriate.
>
>
> >With respect to model comparisons, there is an implicit move to treat different models (sometimes quite different, e.g., word2vec vs. GPT-3.5) as analogous to different human participants (or populations). It is possible, that only certain models produce conceptual structures which are comparable, and thus, this comparison is problematic. The results in the paper about the lack of consistency across tasks shows that even for a single model, they behave in non-human-like ways. Nonetheless, it could be beneficial to note this in the paper.
>
> Here, we do not intend to treat different models as different populations or participants. Instead, we attempt to see which (if any) model can capture the behavioral patterns expressed by a population of human participants. As the reviewer notes, some models better capture conceptual structure than others (GPT-3 is better than FLAN is better than CLIP), but GPT-3 feature listing and GPT-3 triplets, for example, are not as coherent with each other as human feature listing is with human triplets. We take this as evidence that models lack conceptual coherence. Ideally, we agree that it would be a powerful result if, say, GPT3 triplet features strongly cohered with FLAN-XXL triplet features, but we only present the data here for completeness rather than to make a claim about cohere between models.
>
> > Have you tried running the Dutch experiments with GPT-3? It is unclear exactly what languages the model has been exposed to, but it might be an additional point of comparison to see how consistent measures derived from the same model in other languages are (note, I understand that this isn’t possible for some of the other models investigated, so I understand why it isn’t included in the paper).
>
> We think that this is an excellent suggestion and thank the reviewer for it. LLMs seem to have multilingual capabilities[4] and we believe that future work should strive to run precisely the same experiments in different languages on the LLMs as well as construct new human benchmark datasets to validate the model results. It would be interesting to see if conceptual structure coheres more or less for some languages vs. others (e.g., high vs. low resource languages, same vs. different family of languages). The authors of this paper do not speak Dutch unfortunately so it is out of scope for us to run this study in the short term, but if the paper is accepted we will aim to note the strong need for such experiments in the Conclusions section.
>
>
> References:
>
> [1] Newton, P. M., & Xiromeriti, M. (2023). ChatGPT Performance on MCQ Exams in Higher Education. A Pragmatic Scoping Review.
> [2]Kung, T. H., Cheatham, M., Medenilla, A., Sillos, C., De Leon, L., Elepaño, C., ... & Tseng, V. (2023). Performance of ChatGPT on USMLE: Potential for AI-assisted medical education using large language models. PLoS digital health, 2(2), e0000198.
> [3]Choi, J. H., Hickman, K. E., Monahan, A., & Schwarcz, D. (2023). Chatgpt goes to law school. Available at SSRN.
> [4]Armengol-Estapé, J., Bonet, O. D. G., & Melero, M. (2021). On the Multilingual Capabilities of Very Large-Scale English Language Models. arXiv preprint arXiv:2108.13349.

---

### Official Review · Reviewer_DWHV · 2023-08-05

**Soundness:** 4

**Excitement:**

4: Strong: This paper deepens the understanding of some phenomenon or lowers the barriers to an existing research direction.

**Paper Topic And Main Contributions:**

This paper probes the conceptual representations in a class of large language model using cognitive psychology methods designed to assess the similarity between concepts; results are compared to outcomes from the same methods applied to human participants. Results showing some similarities, but striking differences, are relevant for work pushing forward on the strengths and weaknesses of large language models as models for human cognition.

**Questions For The Authors:**

- The diversity of tasks and populations is a strength, but why were these particular datasets chosen (convenience?) Clarifying the motivation here may be helpful, but this is a minor point

**Reasons To Accept:**

- Interesting application of methods from cognitive psychology and research on concepts to large language models
- Use of multiple human datasets spanning tasks, languages, and participant groups is a strong way to demonstrate similarity of human conceptual structure against variable experimental settings.
- Direct comparison between LLM and human data is valuable
- Testing multiple LLMs helps to support generalizability of findings

**Reasons To Reject:**

- Ad hoc choices are, by necessity, used to adapt the human-studies for use with the large language models (e.g. how features were transcribed, how feature verification was implemented etc.) The work might be more impactful if combined with a robustness analysis of the importance of specific analytic choices (this is a small point; time is finite.)

**Reproducibility:**

4: Could mostly reproduce the results, but there may be some variation because of sample variance or minor variations in their interpretation of the protocol or method.

**Reviewer Confidence:**

3: Pretty sure, but there's a chance I missed something. Although I have a good feel for this area in general, I did not carefully check the paper's details, e.g., the math, experimental design, or novelty.

---

> ### Author Rebuttal · Authors · 2023-08-29
>
> We appreciate the reviewer taking the time to read our paper and for their positive comments! We address the concerns raised below -
>
> >Ad hoc choices are, by necessity, used to adapt the human-studies for use with the large language models (e.g. how features were transcribed, how feature verification was implemented etc.) The work might be more impactful if combined with a robustness analysis of the importance of specific analytic choices (this is a small point; time is finite.)
>
> We thank the reviewer for raising this pertinent point. Firstly, our main aim was to implement all tasks (feature listing, feature verification, triplet judgements, pairwise judgments) in the LLMs as similarly as they were conducted in human participants to provide a fair comparison. For LLMs and humans, we asked the agent to list all the features that they knew to be true for a given concept.
> These features were all compiled into a single concept X feature matrix, and the feature verification task was also conducted in a parallel manner between LLMs and humans.
> Here are the specific prompts that were used with Humans and LLMs for the various tasks.
> | | Humans | LLMs |
> |- |-|-|
> | Feature generation | This bundle contains up to 10 sheets with a word written on top of the page.  We would like you to write down preferably 10 features underneath the word.  Try to give different sorts of features, such as, for example, physical or perceptual features (what it looks like, how it smells, how it tastes, ...), functional features (what it is used for, when and where it is used, ...), background information (where it comes from, some historical facts, ...), etc. ((De Deyne et al., 2008)) | List all the properties of {concept1}|
> | Feature verification | The participants were instructed to judge, for every feature-exemplar pair, whether the feature characterizes the exemplar, and to write down a 1 or a 0 in the corresponding matrix entry. (De Deyne et al., 2008)) | In one word, Yes/No : Are {concept1} {property1}  (or) In one word, Yes/No : Do {concept1} have {property1}/ |
> | Triplet task| Which of the bottom two words is more similar to the word at the top? | Answer using only one word - {concept1} or {concept2} and not {anchor}. Which is more similar in meaning to {anchor}?"|
> | Pairwise similarity| On a scale of 1-7, how similar is a {concept1} to a {concept2}? | Answer with only one number from 1 to 7, considering 1 as 'extremely dissimilar', 2 as 'very dissimilar', 3 as 'likely dissimilar', 4 as 'neutral', 5 as 'likely similar', 6 as 'very similar', and 7 as 'extremely similar': How similar is {concept1} and {concept2}? |
>
> However, the reviewer raises a fair point — LLMs are not humans and might be more or less performant with slightly different prompting strategies. This would, in our view, support the central thesis of the paper that conceptual structure does not cohere in LLMs. But to address the reviewer’s concerns, we implemented chain-of-thought reasoning [1], a now well-known prompting strategy shown to improve LLM reasoning capabilities. We found that this led to slightly higher conceptual coherence for only some of the models but still nothing comparable to human conceptual robustness. We will include these results in the supplementary materials.
>
> References:
> [1] Wei, J., Wang, X., Schuurmans, D., Bosma, M., Xia, F., Chi, E., ... & Zhou, D. (2022). Chain-of-thought prompting elicits reasoning in large language models. Advances in Neural Information Processing Systems, 35, 24824-24837.
>
> >The diversity of tasks and populations is a strength, but why were these particular datasets chosen (convenience?) Clarifying the motivation here may be helpful, but this is a minor point.
>
> Thank you for this question! Briefly, the Leuven semantic norms dataset is the only dataset, to our knowledge, that includes a feature verification phase after feature generation. Amongst the subset of concepts we included, they were chosen to span semantic domains (living and nonliving) and levels of granularity. Future studies can work on expanding our approach to other semantic norm datasets.

---

### Meta-Review · Area_Chair_pLno · 2023-09-14

**Recommendation:** 5

**Metareview:**

This work compares conceptual structure in humans and language models. The research question is how consistent is the conceptual strcuture across different tasks for humans vs for models. The findings are that conceptual structure is more consistent across people than it is across models.

Overall, the reviewers unanimously agree that the work is sound and the contributions are valuable and would be exciting to the EMNLP audience. Specifically, the reviewers praise the comprehensive human experimental conditions spanning different datasets, languages, tasks, the clarity of the arguments, and the timely nature of the comparison between human behavior and large language models. Reviewers also point out a small number of opportunities for improvement, including motivating experimental choices made in the human experiments and an addition of an agreement analysis among the human annotators, which the authors responded to and reported to be high. This information should be included in a revision.

---

### Decision · Program_Chairs · 2023-10-07

**Decision:**

Accept-Main

**Comment:**

This work compares conceptual structure in humans and language models. The research question is how consistent is the conceptual strcuture across different tasks for humans vs for models. The findings are that conceptual structure is more consistent across people than it is across models.

Overall, the reviewers unanimously agree that the work is sound and the contributions are valuable and would be exciting to the EMNLP audience. Specifically, the reviewers praise the comprehensive human experimental conditions spanning different datasets, languages, tasks, the clarity of the arguments, and the timely nature of the comparison between human behavior and large language models. Reviewers also point out a small number of opportunities for improvement, including motivating experimental choices made in the human experiments and an addition of an agreement analysis among the human annotators, which the authors responded to and reported to be high. This information should be included in a revision.